



**A new global oceanic multi-model net primary productivity data product**
Thomas J. Ryan-Keogh[1*], Sandy J. Thomalla[1,2], Nicolette Chang[1], Tumelo Moalusi[1,3]
[1] Southern Ocean Carbon-Climate Observatory, CSIR, Cape Town, South Africa
[2] Marine and Antarctic Research Centre for Innovation and Sustainability, Department of Oceanography,
University of Cape Town, Cape Town, South Africa
[3] Global Change Institute, University of Witwatersrand, Johannesburg, South Africa
*Corresponding Author: Thomas Ryan-Keogh; tryankeogh@csir.co.za
**Abstract.** Net primary production of the oceans contributes approximately half of the total global net
primary production and long-term observational records are required to assess any climate driven changes.
The Ocean Colour Climate Change Initiative (OC-CCI) has proven to be robust, whilst also being one of
the longest records of ocean colour. However, to date only one primary production algorithm has been
applied to this data product with other algorithms typically applied to single sensor missions. The data
product presented here addresses this issue by applying five algorithms to the OC-CCI data product, which
allows the user to interrogate the range of distribution across multiple models and to identify consensus or
outliers for their specific region of interest. Outputs are compared to single sensor data missions
highlighting good overall global agreement, with some small regional discrepancies. Inter-model
assessments address the source of these discrepancies, highlighting the choice of the mixed layer data
product as a vital component for accurate primary production estimates.
**1 Introduction**
Phytoplankton primary production and associated seasonal blooms play an important role in the carbon
cycle, being responsible for approximately 50% of total global net primary production (NPP) (Lurin, 1994;
Longhurst et al., 1995; Field et al., 1998; Carr et al., 2006; Buitenhuis et al., 2013). Global NPP estimates
are in the order of 50 Gt C per year (Longhurst et al., 1995; Field et al., 1998; Carr et al., 2006; Buitenhuis
et al., 2013; Antoine et al., 1996; Silsbe et al., 2016; Johnson and Bif, 2021). When this organic carbon is
sequestered to the ocean interior via the biological carbon pump (BCP) it can offset the flux of upwelled
pre-industrial dissolved inorganic carbon (Mikaloff Fletcher et al., 2007; Gruber et al., 2009). In that sense,
in the contemporary period, it does not play a significant role in the ocean uptake of anthropogenic carbon
dioxide ($CO_2$). However, the magnitude of the BCP is predicted to change in response to global climate



change, altering the ocean's ability to store carbon and hence atmospheric levels of $CO_2$ (Henson et al.,
2011; Bopp et al., 2013; Boyd et al., 2015; Tagliabue et al., 2021). In that sense, any natural or
anthropogenic perturbations to the strength and efficiency of the BCP have the potential to drive important
feedbacks on global climate change and thus need to be considered for a comprehensive understanding of
the trajectory of the ocean carbon sink. Recent studies have estimated that global NPP is indeed changing,
with declines ranging from 0.6 to 13% (Gregg and Rousseaux, 2019; Polovina et al., 2011; Chavez et al.,
2010; Behrenfeld et al., 2006) and increases of up to 2% (Saba et al., 2010). These changes are of concern
given that alterations in the contribution that the BCP plays in offsetting upwelling of DIC will impact the
net uptake of anthropogenic $CO_2$ (Henson et al., 2011). NPP also plays an important role in supporting
ecosystem function by sustaining biodiversity and the transfer of carbon, energy, and nutrients through
pelagic and benthic food webs. As such, any changes to the amount of bulk carbon being produced is likely
to impact the amount of carbon available for transfer to higher trophic levels via the marine food web with
implications for ecosystem health and fisheries success. It is the seasonal cycle that sets much of the
environmental variability in the factors that drive NPP, and it is the dominant mode of variability that
couples the physical mechanisms of climate forcing to ecosystem response in production, diversity and
carbon export (Monteiro et al., 2011). As such, understanding the seasonal evolution of NPP can provide a
sensitive index of climate variability through its dependence on physical processes that transport nutrients
and control the exposure of phytoplankton to sunlight (Summer and Lengfeller, 2008; Henson et al., 2009).
It is with this in mind that we seek to provide a data product that can be used to understand the extent to
which the seasonal characteristics of NPP are being modified by environmental conditions over sufficiently
long time periods. NPP has already been highlighted as a better indicator of environmental change and
disturbances in comparison to chlorophyll-a (Tilstone et al., 2023), highlighting its suitability for ecosystem
assessment of tipping points and abrupt change.

Phytoplankton NPP is strongly influenced by the physico-chemical conditions of the ocean, including light,
temperature, macronutrient and micronutrient concentrations. Climate change has already begun to elicit
widespread changes to these conditions, for example increases in temperature and heat content, increased
sea ice melt and enhanced precipitation all contribute to alterations of oceanic density and the subsequent
nutrient supply into the euphotic zone (Field et al., 2014; Rhein et al., 2013). Being able to understand how
these climate driven changes in the physico-chemical environment impact phytoplankton NPP is key to
addressing one of the most important scientific and policy challenges of the 21[st] century, namely being able
to predict long term trends in the ocean carbon - climate system. This challenge is exacerbated by the
sparsity of NPP data and a lack of continuous or regular in situ measurements for long enough periods to
address multi decadal changes associated with climate forcing.




Satellite based remote sensing of ocean colour is the only observational capability that can provide synoptic
views of upper ocean phytoplankton characteristics at high spatial and temporal resolution (~1km, ~daily)
and high temporal extent (global scales, years to decades). In many cases these are the only systematic
observations available for chronically under-sampled marine systems such as the Southern Ocean.
Empirical expressions of estimating NPP are built around long recognised dependencies between
phytoplankton biomass and environmental conditions (e.g., temperature, light and nutrients), with a
succinct review available in Westberry et al. (2023). The vertically Generalized Production Model (VGPM)
(Eppley, 1972; Behrenfeld and Falkowski, 1997) is a simpler satellite NPP model that relies on the
relationship between chlorophyll and temperature derived growth rates with no explicit spectral, temporal,
or vertical resolution. The Carbon-based Production Models (CbPM; Behrenfeld et al., 2005; Westberry et
al., 2008) rely on particulate backscattering estimates of phytoplankton carbon as a biomass indicator
instead of chlorophyll. This approach allows for some of the variability in chlorophyll to be attributed to
physiological adjustments to light and nutrients (e.g., photoacclimation), independent of changes in NPP.
The more recent CAFE model (Silsbe et al., 2016) builds upon this approach but in addition incorporates
the influence of non-algal absorption on the attenuation of the underwater light field, which if not accounted
for has a tendency to overestimate NPP (notably in coastal waters). Recently, considerable effort has been
invested to provide one of the longest records of ocean colour by merging data and correcting inter-sensor
biases from multiple ocean colour satellite sensors (Sathyendranath et al., 2019a), known as the Ocean
Colour Climate Change Initiative (OC-CCI). This time series of 25 years (as of 2023) has already been
utilised to provide estimates of trends in global NPP (Kulk et al., 2020), with results showing that trends in
NPP were linked to trends in chlorophyll-$a$ and related to changes in the physico-chemical conditions of
the water column from inter-annual and multi-decadal climate oscillations. However, this study only
investigated one NPP algorithm as opposed to using a suite of different algorithms with varying sensitivities
to specific processes, as is done for the assessments of predicted change from earth system models in the
coupled model intercomparison project (CMIP).

Given the importance of NPP for assessing carbon budgets, ecosystem health and environmental change it
is becoming increasingly clear that users require easy access to appropriate data products. Unfortunately,
the global NPP algorithm applied to OC-CCI by Kulk et al. (Kulk et al., 2020) is not available for download
on the OC-CCI server. Although an NPP data product is available from Copernicus Marine Services, this
is only applied to the temporally limited GlobColour data product and similarly is only available for a single
NPP algorithm (Antoine and Morel, 1996). The most comprehensive suite of NPP algorithms is provided
by the Ocean Productivity website (http://sites.science.oregonstate.edu/ocean.productivity/custom.php),





however these are also only applied to single sensor missions (SeaWIFS, MODIS, VIIRS) thus restricting
time periods of interest and preventing any longer-term assessments of change. Furthermore, it is difficult
for the user to ascertain exactly which ancillary data products (i.e., MLD criterion, nitracline) were being
used in the final empirical derivation of NPP.

Here we present a new ocean colour data product that incorporates 5 NPP algorithms applied to the 25-year
merged sensor OC-CCI time series. This multi-model data product provides a range of estimates of global
NPP from 1998 to 2022 at both 8-day and monthly resolution and at a spatial coverage of 25 km. The
distribution of the models are assessed across different oceanic biomes and long term observatory sites to
highlight either consensus or outliers. The outputs of these algorithms are assessed for any biases or
differences in comparison to the original outputs from single sensor missions and intra-algorithm
differences for the multi-sensor satellite record.

**2 Materials and Methods**

25 years of ocean colour data from 1998 – 2022 were downloaded from the OC-CCI server (8-day; 4 km;
v6.0; Sathyendranath et al., 2019), including chlorophyll *a* concentration (chl-*a*; mg m$^{-3}$), backscatter at 443
nm ($b_{bp}$; m$^{-1}$), the diffuse attenuation coefficient at 490 nm ($K_d$ 490; m$^{-1}$), the phytoplankton absorption
coefficient at 443 nm ($a_{ph}$; m$^{-1}$) and the detrital absorption coefficient at 443 nm ($a_{dg}$; m$^{-1}$). As the OC-CCI
server does not contain the spectral slope of $b_{bp}$ ($\eta$; m$^{-1}$ nm$^{-1}$), it was calculated following equation 1 from
Pitarch et al. (2019) using remote sensing reflectance ($R_{rs}$) at 443 nm and 560 nm. Daily integrated
photosynthetically active radiation (PAR; mol photons m$^{-2}$ d$^{-1}$) data were downloaded from Glob-Colour
(http://www.globcolour.info/). Sea surface temperature (SST; °C) data were downloaded from the Group
for High Resolution Sea Surface Temperature (GHRSST; https://www.ghrsst.org/). The Hadley EN4.2.2
gridded temperature and salinity profiles (Good et al., 2013) were converted to density ($\sigma$; kg m$^{-3}$) to derive
mixed layer depth (MLD; m) using the density thresholds of 0.03 kg m$^{-3}$ (de Boyer Montégut et al., 2004)
and 0.125 kg m$^{-3}$. Additional data for MLD were retrieved from HYCOM
(https://www.hycom.org/data/glba0pt08), for both density criteria (downloaded from
http://sites.science.oregonstate.edu/ocean.productivity/).

For the primary analysis of the paper the outputs using the Hadley $\Delta\sigma_{10m} = 0.030$ kg m$^{-3}$ MLD data product
was used (Ryan-Keogh, 2023d). The reason for this choice were concerns around the accuracy of the
HYCOM MLD data product to best represent in situ conditions. A trend analysis performed on all MLD
products and criterion (Figure A1) revealed distinct directional differences in the trends of Hadley versus



HYCOM, with the Hadley MLD product the only one to best represent the global MLD trends as outlined
in Sallée et al. (2021). However, the outputs using Hadley $\Delta\sigma_{10m}$ = 0.125 kg m$^{-3}$ (Ryan-Keogh, 2023a),
HYCOM $\Delta\sigma_{10m}$ = 0.030 kg m$^{-3}$ (Ryan-Keogh, 2023b) and HYCOM $\Delta\sigma_{10m}$ = 0.125 kg m$^{-3}$ (Ryan-Keogh,
2023c) are all available.

The nitracline depth was defined as the depth at which nitrate and nitrite was equal to 0.5 μM (Westberry
et al., 2008), using the monthly data from the World Ocean Atlas 2018 (WOA18; (Garcia et al., 2019). The
total backscattering of pure seawater ($b_{bw}$; m$^{-1}$) was derived as a function of SST and salinity following
Zhang and Hu (2009), using monthly salinity data from WOA18 averaged for the top 20 m.

All data were regridded onto a regular grid of 25 km spatial resolution, using bilinear interpolation using
the xESMF Python package (Zhuang, 2018), at 8-day temporal resolution. The remaining gaps were filled
by applying a linear interpolation scheme in sequential steps of longitude, latitude and time (Racault et al.,
2014) using a three-point window. If one of the points bordering the gap along the indicated axis was invalid
it was omitted from the calculation, whilst if two surrounding points were invalid then the gap was not
filled. Finally, the data were smoothed by applying a moving average filter of the previous and next
timestep. For more details on this method see Salgado-Hernanz et al. (2019).

NPP (mg C m$^{-2}$ d$^{-1}$) was calculated using 5 different algorithms, the 'Eppley-VGPM' model (Eppley, 1972),
the 'Behrenfeld-VGPM' model (Behrenfeld and Falkowski, 1997), the 'Behrenfeld-CbPM' model
(Behrenfeld et al., 2005), the 'Westberry-CbPM' model (Westberry et al., 2008) and the 'Silsbe-CAFE'
model (Silsbe et al., 2016). Both Eppley-VGPM and Behrenfeld-VGPM models are chlorophyll based
production models with a temperature-dependent derivation of photosynthetic efficiencies. The Behrenfeld-
CbPM and Westberry-CbPM models are based upon deriving carbon biomass from backscatter coefficients
and growth rates from chlorophyll-to-carbon ratios, with the Westberry-CbPM being spectrally resolved
across 9 wavelengths. The Silsbe-CAFE model is an absorption based model that is spectrally resolved
across 21 wavelengths, whilst also being resolved across the diel cycle from sunrise to sunset. For more
details on which parameters are required for each model please see Table 1.

| Chl-$a$ | PAR | $b_{bp}$ | $a_{ph}$ | $a_{dg}$ | $K_d$ | $\eta$ | $b_{bw}$ | MLD | SST | Nitracline | SSS |
|---|---|---|---|---|---|---|---|---|---|---|---|
|  |  |  |  |  |  |  |  |  |  |  |  |





| | | | | | | | | | | | | |
|---|---|---|---|---|---|---|---|---|---|---|---|---|
| *Eppley-VGPM* | ✓ | ✓ | × | × | × | × | × | × | × | ✓ | × | × |
| *Behrenfeld-VGPM* | ✓ | ✓ | × | × | × | × | × | × | × | ✓ | × | × |
| *Behrenfeld-CbPM* | ✓ | ✓ | ✓ | × | × | ✓ | × | × | ✓ | × | × | × |
| *Westberry-CbPM* | ✓ | ✓ | ✓ | × | × | ✓ | × | × | ✓ | × | ✓ | × |
| *Silsbe-CAFE* | ✓ | ✓ | ✓ | ✓ | ✓ | ✓ | ✓ | ✓ | ✓ | ✓ | × | ✓ |

Table 1: Data variables, including chlorophyll-a (Chl-$a$; mg m$^{-3}$), photosynthetically active radiation (PAR; mol photons m$^{-2}$ d$^{-1}$), backscatter at 443 nm ($b_{bp}$; m$^{-1}$), phytoplankton absorption at 443 nm ($a_{ph}$; m$^{-1}$), detrital absorption at 443 nm ($a_{dg}$; m$^{-1}$), diffuse attenuation coefficient at 490 nm ($K_d$; m$^{-1}$), the spectral slope of backscatter ($\eta$; m$^{-1}$ nm$^{-1}$), the backscatter of pure water ($b_{bw}$; m$^{-1}$), mixed layer depth (MLD; m), sea surface temperature (SST; °C), nitracline depth (m) and sea surface salinity (SSS), used in the derivation of net primary production using 5 models including the Eppley-VGPM, Behrenfeld-VGPM, Behrenfeld-CbPM, Westberry-CbPM and Silsbe-CAFE.

For presentation purposes the global data were separated into biomes using the classification from Fay & McKinley (2014), while long-term observatories were selected as the Bermuda Atlantic Time Series (30.7-32.7°N, 59.2-61.2°W), the Hawaii Oceanic Time Series (21.8-23.8°N, 157-159°W), the Southern Ocean Time Series (46.0-48.0°S, 139-141°E) and the Porcupine Abyssal Plain observatory (48-50°N, 15.5-17.5°W).

For comparison to the OC-CCI outputs presented here, monthly NPP data of Eppley-VGPM, Behrenfeld-VGPM, Westberry-CbPM and Silsbe-CAFE were downloaded from the Ocean Productivity website (http://sites.science.oregonstate.edu/ocean.productivity/) for SeaWIFS (1998 - 2007) and MODIS (2003 - 2019). Unfortunately, the NPP data for the Behrenfeld-CbPM is no longer available as it has been superseded by the Westberry-CbPM NPP data. Pearson's correlation coefficients ($R^2$) were calculated between the SeaWIFS/MODIS derived NPP and the OC-CCI derived NPP.

**3 Results & Discussion**



Comparing intra-model climatological means

The climatological means of each NPP model show a large degree of spatial heterogeneity, with higher
values associated with western boundary currents and at the equator (Figure 1). The temperature based
Eppley-VGPM and Behrenfeld-VGPM models (Figure 1a,b) show good agreement in terms of their ranges
and means (Table 2), but there are large differences particularly in the North Atlantic and the Arabian Sea
and equatorial Pacific. The carbon based Behrenfeld-CbPM and Westberry-CbPM models (Figure 1c,d)
show very good agreement in terms of their climatological means although discrepancies are nonetheless
evident (e.g. higher NPP in the Southern Ocean and North Atlantic in the Behrenfeld-CbPM and higher
NPP in the equatorial region in the Westberry-CbPM). The absorption based Silsbe-CAFE model (Figure
1e) has a much smaller range across the global ocean. A map of the coefficient of variation (CV =
$\sigma$NPP/<NPP>; Figure 2a) shows the highest values (depicting agreement between models) in the high
latitudes and in coastal regions. Unlike the comparison in Westberry et al. (2023) (which included
Behrenfeld-VGPM, Westberry-CbPM and Silsbe-CAFE applied to MODIS data from 2003 to 2019), we
do not find lower CV values to be specifically associated with highly productive waters, nor do we find a
similar distribution for very high CV values. The Silsbe-CAFE model has the most peaked probability
distributions (PDF) of all the models (Figure 2b) with a narrow range, which is similar to that reported in
Westberry et al. (2023). The other models show a much lower peak and broader range with the two CbPM
models centred around a lower median distribution of NPP (more similar to that of Silsbe-CAFE) than the
slightly higher median NPP of the two VGPM models. When we examine the cumulative distributions
(CDF) of each model (Figure 2c), the medians were an order of magnitude higher in the Eppley-VGPM
(1019.5 mg C $m^{-2}$ $d^{-1}$) and Behrenfeld-VGPM (1206.6 mg C $m^{-2}$ $d^{-1}$) in comparison to the Behrenfeld-CbPM
(298.2 mg C $m^{-2}$ $d^{-1}$), Westberry-CbPM (531.1 mg C $m^{-2}$ $d^{-1}$) and Silsbe-CAFE (495.5 mg C $m^{-2}$ $d^{-1}$). Whilst
the median values for both Westberry-CbPM and Silsbe-CAFE are similar to those reported in Westberry
et al. (2023), the Behrenfeld-VGPM values are much higher than what was previously reported (332 mg C
$m^{-2}$ $d^{-1}$), which is not necessarily surprising when considering that different SST, PAR and Chl-a products
are being used in this analysis.

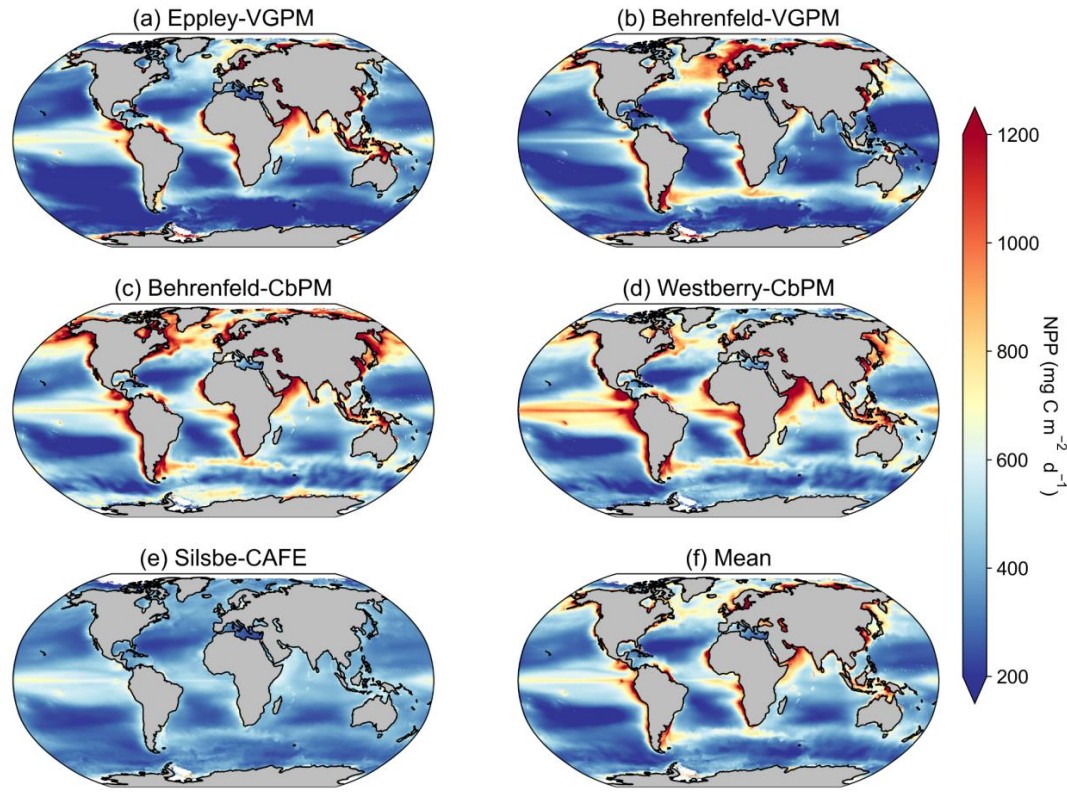

Figure 1: Climatological means of net primary productivity (NPP) for the period of 1998-01-01 to 2022-
12-31 for the (a) Eppley-VGPM, (b) Behrenfeld-VGPM, (c) Behrenfeld-CbPM, (d) Westberry-CbPM, (e)
Silsbe-CAFE model and (f) the mean of all models.

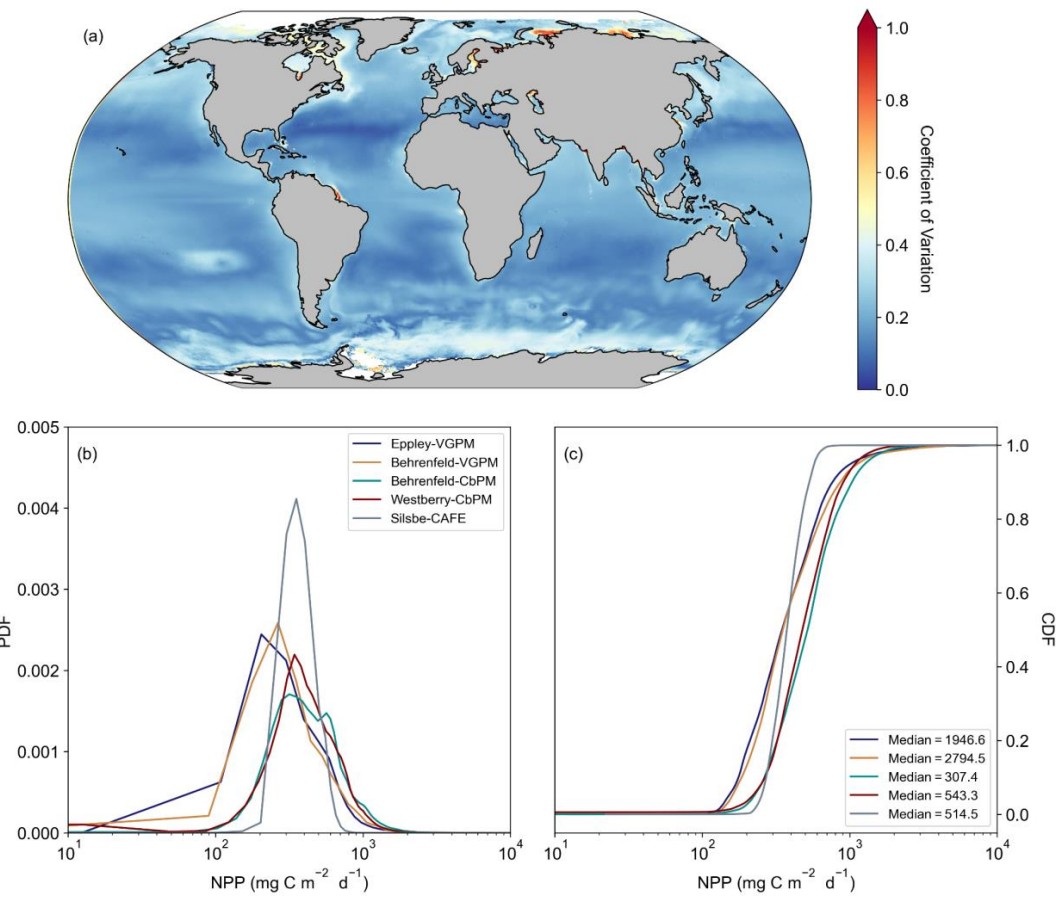


Figure 2: The distribution of the model net primary production (NPP) values. (a) the coefficient of variation, calculated as the inter-model standard deviation normalised to the inter-model mean. (b) Probability distributions (PDF) of the climatological mean NPP for each of the models. (c) Cumulative distributions of the climatological mean NPP for each of the models.


| | MLD criterion | Min | Max | Mean±Stdev | Median | IQR | Global NPP |
|---|---|---|---|---|---|---|---|
| *Eppley-VGPM* | n/a | 12.7 | 18941.2 | 457.8±464.4 | 347.4 | 311.2 | 68.3±2.8 |
| *Behrenfeld-VGPM* | n/a | 11.8 | 17289.1 | 485.2±470.7 | 352.0 | 317.8 | 69.2±2.6 |


Earth System
Open Access  Science
Data  Discussions

| | | | | | | | |
|---|---|---|---|---|---|---|---|
| **Behrenfeld-CbPM** | Hadley $\Delta\sigma_{10m}$ = 0.03 kg m$^{-3}$ | $7.4 \times 10^{-8}$ | 6933.1 | 596.7±362.9 | 517.9 | 368.5 | 86.1±2.5 |
| | Hadley $\Delta\sigma_{10m}$ = 0.125 kg m$^{-3}$ | $7.9 \times 10^{-13}$ | 6605.1 | 458.8±293.3 | 387.3 | 308.7 | 69.2±1.8 |
| | HYCOM $\Delta\sigma_{10m}$ = 0.03 kg m$^{-3}$ | $1.7 \times 10^{-11}$ | 22459.4 | 655.3±770.0 | 488.0 | 349.7 | 77.6±6.3 |
| | HYCOM $\Delta\sigma_{10m}$ = 0.125 kg m$^{-3}$ | $2.0 \times 10^{-11}$ | 26208.4 | 743.0±1264.0 | 425.2 | 360.0 | 87.8±7.4 |
| **Westberry-CbPM** | Hadley $\Delta\sigma_{10m}$ = 0.03 kg m$^{-3}$ | $9.7 \times 10^{-29}$ | 7183.2 | 545.6±292.3 | 477.8 | 331.8 | 84.6±2.6 |
| | Hadley $\Delta\sigma_{10m}$ = 0.125 kg m$^{-3}$ | $1.9 \times 10^{-28}$ | 6894.5 | 456.8±279.6 | 380.5 | 318.7 | 73.0±2.3 |
| | HYCOM $\Delta\sigma_{10m}$ = 0.03 kg m$^{-3}$ | $3.2 \times 10^{-12}$ | 25505.9 | 506.7±267.0 | 451.6 | 310.4 | 78.0±3.3 |
| | HYCOM $\Delta\sigma_{10m}$ = 0.125 kg m$^{-3}$ | $9.1 \times 10^{-157}$ | 135462.7 | 454.1±323.8 | 390.8 | 302.8 | 70.1±3.5 |
| **Silsbe-CAFE** | Hadley $\Delta\sigma_{10m}$ = 0.03 kg m$^{-3}$ | 22.1 | 1193.2 | 388.8±100.5 | 374.7 | 137.4 | 59.3±3.9 |
| | Hadley $\Delta\sigma_{10m}$ = 0.125 kg m$^{-3}$ | 22.1 | 1193.2 | 383.1±100.9 | 365.2 | 141.1 | 58.9±3.8 |
| | HYCOM $\Delta\sigma_{10m}$ = 0.03 kg m$^{-3}$ | 22.3 | 1204.1 | 386.6±99.6 | 371.0 | 138.3 | 59.2±3.9 |
| | HYCOM $\Delta\sigma_{10m}$ = 0.125 kg m$^{-3}$ | 17.9 | 1193.2 | 378.3±102.5 | 361.9 | 140.9 | 58.3±3.8 |

Table 2: The climatological global minimum, maximum, mean±standard deviation, median and
interquartile range (IQR: 75th - 25th) for each net primary production model. Included is the sum of the



global NPP (Pg C yr⁻¹) from each model (averaged for each year from 1998 to 2022, including the standard
deviation), including the different MLD criterion used (where n/a means not applicable).

Investigating the difference in climatological means between each model and the ensemble model mean
highlights the regional distribution of positive and negative biases relative to the ensemble model mean
(Figure A2). For example, the two VGPM models show an opposite distribution in their relative differences
with Behrenfeld-VGPM being higher in the North-Atlantic, Arctic and Antarctic Circumpolar Current
(ACC) regions while Eppley-VGPM is higher in the equatorial region. Both CbPM models show a tendency
to overestimate NPP compared to other models except in the Arctic where the Westberry-VBPM is instead
lower than the ensemble model mean. Interestingly, although the climatological mean of the Silsbe-CAFE
appears lower than all other models (Figure 1) this is not globally consistent when expressed as a difference
which instead highlights that the Silsbe-CAFE overestimates NPP relative to other models in the
oligotrophic gyres and ACC region.

Finally, if we compare global oceanic NPP from the models with previous IPCC estimates of 50 Pg C m⁻²
yr⁻¹, only the Silsbe-CAFE model has a similar range in NPP (58.9 - 59.3 Pg C m⁻² yr⁻¹), whereas the ranges
of all the other models are much higher (68.3 - 87.8 Pg C m⁻² yr⁻¹), with some estimates higher than
previously reported (32.0 - 70.7 Pg C m⁻² yr⁻¹; Buitenhuis et al., 2013; Sathyendranath et al., 2019b).

Interrogating spatio-temporal patterns of NPP Data Products

Fay and McKinley (2014) classified the global ocean into 17 biomes (Figure 3) according to distinct
biological (chl-*a* concentrations) and physical characteristics (SST, MLD and ice fraction). Splitting the
NPP data according to these biomes allows a regional comparison of inter model differences and
similarities. The annual model means of each NPP product range from a minimum value of 207.85±38.67
mg C m⁻² d⁻¹ in the Southern Ocean subpolar seasonally stratified (SO SPSS) biomes (Figure 3s) to a
maximum value of 652.21±135.06 mg C m⁻² d⁻¹ in the East Pacific equatorial biome (PEQU E) biome
(Figure 3g). When globally averaged (Figure 3s) the models appear to agree very well in their annual
climatologies of NPP, however when interrogated on a per biome basis, some discrepancies emerge. For
example, although there is particular good agreement in NPP in the oligotrophic gyres (Figure 3e,h,l,n),
large intra-model differences are particularly evident in the equatorial biomes (Figure 3f,g,m) and the high
latitude Atlantic and Pacific (Figure 3b,c,i,j). In some biomes there is also a tendency for models to merge
or diverge over time. For example, there is a large inter model spread in the early 2000's in the North
Atlantic and Southern Ocean ICE biomes (Figure 3 i, r), which narrows over time, while the opposite is





apparent in the North Atlantic Subpolar seasonally stratified biome (NA SPSS) biome (Figure 3j). Also
worth noting are regions where all models agree except one, for example the comparatively lower NPP for
the Behrenfeld-VGPM model in the West Pacific equatorial biome (PEQU W) (Figure 3f).




Figure 3: (a) Map of the mean biomes from Fay and McKinley (2014), where white areas represent regions which do not fit into any biome classification. Annual means of net primary productivity (NPP; mg C m$^{-2}$ d$^{-1}$) from the Eppley-VGPM, Behrenfeld-VGPM, Behrenfeld-CbPM, Westberry-CbPM and Silsbe-CAFE model for the (b) North Pacific Ice biome (NP ICE), (c) North Pacific Subpolar seasonally stratified biome (NP SPSS), (d) North Pacific Subtropical seasonally stratified biome (NP STSS), (e) North Pacific Subtropical permanently stratified biome (NP STPS), (f) West Pacific equatorial biome (PEQU W), (g) East Pacific equatorial biome (PEQU E), (h) South Pacific Subtropical permanently stratified biome (SP STPS), (i) North Atlantic ice biome (NA ICE), (j) North Atlantic Subpolar seasonally stratified biome (NA SPSS), (k) North Atlantic Subtropical seasonally stratified biome (NA STSS), (l) North Atlantic Subtropical permanently stratified biome (NA STPS), (m) Equatorial Atlantic biome (AEQU), (n) South Atlantic Subtropical permanently stratified biome (SA STPS), (o) Indian Subtropical permanently stratified biome (IND STPS), (p) South Ocean Subtropical seasonally stratified biome (SO STSS), (q) Southern Ocean Subpolar seasonally stratified biome (SO SPSS), (r) Southern Ocean ice biome (SO ICE) and (s) the global ocean.

In the next model comparison, we combine biomes into three regions; the northern high latitude, equatorial and southern high latitude to examine the seasonal cycle in NPP across the five models. Here, inter-model differences become even more pronounced in terms of their minima, maxima and phenology of the seasonal cycle (Figure 4). In the northern hemisphere biomes (Figure 4a; North Pacific and North Atlantic ice, subpolar seasonally stratified and subtropical seasonally stratified biomes) there is a large range of variability in maximum NPP, with the Behrenfeld-VGPM and Behrenfeld-CbPM exhibiting the highest peak values (919.60 and 936.59 mg C m$^{-2}$ d$^{-1}$, respectively) and the Silsbe-CAFE model exhibiting the lowest peak value (507.41 mg C m$^{-2}$ d$^{-1}$). The timing of the peaks are also offset with the earliest peak occurring in the Eppley-VGPM, Behrenfeld-VGPM and Silsbe-CAFE models at the start of June while the Behrenfeld-CbPM and Westberry-CbPM models puts the timing of the peak a few weeks later in mid-June. The southern hemisphere biomes (Figure 4c; Southern Ocean ice, subpolar seasonally stratified and subtropical seasonally stratified biomes) similarly express a large range in amplitude of the seasonal peak across all models, with both CbPM models exhibiting the highest values (758.15 and 649.42 mg C m$^{-2}$ d$^{-1}$, respectively) whereas the Eppley-VGPM exhibits the lowest peak value (379.62 mg C m$^{-2}$ d$^{-1}$). The timing of the peak is similar for Behrenfeld-CbPM, Westberry-CbPM and Silsbe-CAFE in January with the Eppley-VGPM and Behrenfeld-VGPM models placing the bloom peak earlier in December. The low latitude and equatorial biomes (Figure 4b; North & South Pacific subtropical permanently stratified, North & South Atlantic subtropical permanently stratified, Indian subtropical permanently stratified, Atlantic and Pacific equatorial biomes) do not exhibit any clear seasonal cycle and have a lower range of variability



across all the models. The range of divergence is more similar to that of the seasonal troughs of NPP in the
Northern and Southern high latitude regions, although rates of NPP are not as low (mean for all models for
the time series = 412.85±69.86 mg C m$^{-2}$ d$^{-1}$).

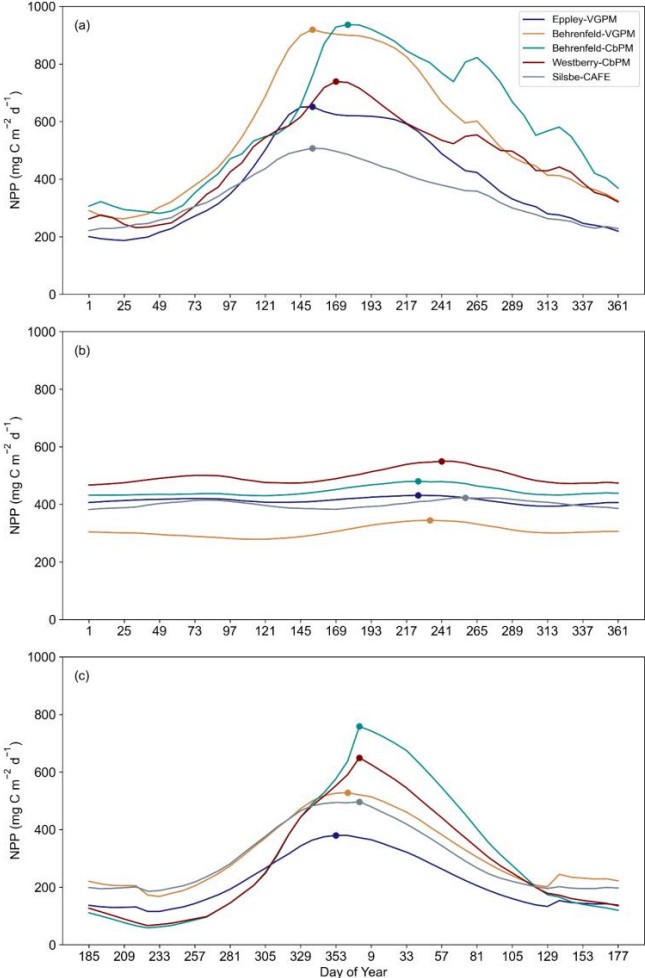


Figure 4: The seasonal cycle of net primary productivity (NPP; mg C m$^{-2}$ d$^{-1}$) from the Eppley-VGPM,
Behrenfeld-VGPM, Behrenfeld-CbPM, Westberry-CbPM and Silsbe-CAFE models for (a) the northern
high latitude regions (NA ICE, NP ICE, NA SPSS, NP SPSS, NA STSS and NP STSS), (b) the equatorial
and low latitude regions (AEQU, PEQU E, PEQU W, IND STPS, NA STPS, SA STPS, NP STPS, SP
STPS) and (c) the southern high latitude regions (SO ICE, SO SPSS and SO STSS). Data is averaged across
the time period 1998 – 2022. Please note that for panel c the data has been shifted for the peak to appear in
the centre of the plot. The circles represent the timing of the annual maximum.




We further examined the variability between models by choosing 4 long-term observatory sites; the
porcupine abyssal plain observatory (PAP; Figure 5a), the Bermuda Atlantic Time Series (BATS; Figure
5b), the Hawaii Oceanic Time Series (HOTS; Figure 5c) and the Southern Ocean Time Series (SOTS;
Figure 5d). The BATS site has the lowest range of NPP with the smallest inter-model differences
(305.12±45.54 mg C m$^{-2}$ d$^{-1}$), while HOTS and SOTS express a similar range in NPP (351.86±68.31 &
345.42±76.54 mg C m$^{-2}$ d$^{-1}$, respectively) and the PAP site has the highest range in NPP and greatest inter-
model differences (625.83±190.81 mg C m$^{-2}$ d$^{-1}$).

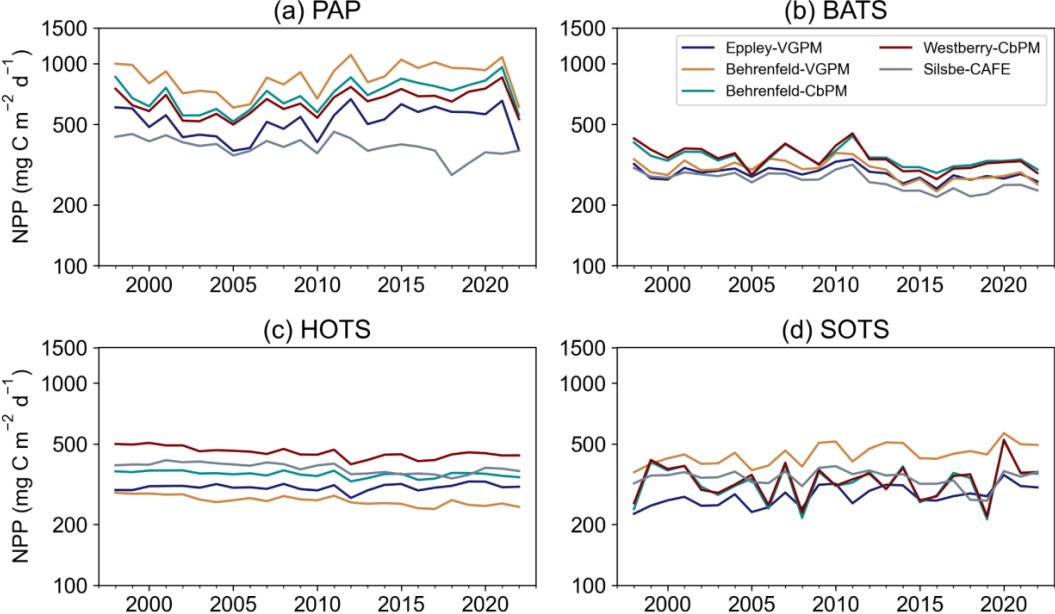


Figure 5: Annual means of net primary productivity (NPP; mg C m$^{-2}$ d$^{-1}$) from the Eppley-VGPM,
Behrenfeld-VGPM, Behrenfeld-CbPM, Westberry-CbPM and Silsbe-CAFE models for (a) the Porcupine
Abyssal Plain (PAP) observatory, (b) the Bermuda Atlantic Time Series (BATS), (c) the Hawaii Oceanic
Time Series (HOTS), and (d) the Southern Ocean Time Series (SOTS).

Comparison with MODIS and SeaWIFS derived NPP

When first designed, these NPP models were originally implemented on both SeaWIFS and MODIS data
products. As such, we are able to compare the new OC-CCI derived NPP for all models presented here with
the original NPP from both SeaWIFS and MODIS that is downloadable from the Ocean Productivity





website (http://sites.science.oregonstate.edu/ocean.productivity/). Spatial correlation maps were
subsequently derived for the Eppley-VGPM, Behrenfeld-VGPM, Westberry-CbPM and Silsbe-CAFE
models using both SeaWIFS and OC-CCI derived NPP for the period of 1998-01-01 to 2007-12-31 (Figure
A3) and the MODIS and OC-CCI derived NPP for the period 2003-01-01 to 2019-12-31 (Figure A4).
Results show very good agreement for Eppley-VGPM (Figure A3a,b; Figure A4a,b) and Behrenfeld-
VGPM (Figure A3c,d; Figure A4c,d) for both SeaWIFS (median $R^2$ = 0.83 and 0.87 respectively) and
MODIS (median $R^2$ = 0.85 and 0.89 respectively) with some lower $R^2$ values evident in the equatorial
region. Correlations were generally poor for the Westberry-CbPM model for both SeaWIFS (median $R^2$ =
0.41) and MODIS (median $R^2$ = 0.51). Correlations against the Silsbe-CAFE model were good at higher
latitudes for both SeaWIFS and MODIS but poor in the equatorial region with the overall correlation being
worse for MODIS (median $R^2$ = 0.65) than for SeaWIFS (median $R^2$ = 0.69). However, the NPP data
products generated from SeaWIFS and MODIS for these respective time periods were derived using the
HYCOM MLD data product and not Hadley (as per the OC-CCI NPP product), which may account for
some of the observed variability and poor correlations. For consistency, we can instead similarly use the
HYCOM MLD with a density criterion of $\Delta\sigma_{10m}$ = 0.030 kg m$^{-3}$ (Figure A5) to derive the OC-CCI NPP
product for comparison with SeaWIFS and MODIS products for the Westberry-CbPM and Silsbe-CAFE
models (which both use MLD as input criteria unlike the VGPM models) (Ryan-Keogh, 2023b). Here we
see an overall improvement in the spatial correlation maps and distribution of $R^2$ which for Westberry-
CbPM increased in both SeaWIFS and MODIS to an $R^2$ = 0.50 and 0.60, respectively, while for the Silsbe-
CAFE model the correlation increased to an $R^2$ = 0.76 and 0.70 (for SeaWIFS and MODIS, respectively).

The reasons for discrepancies between NPP products derived from OC-CCI versus SeaWIFS/MODIS can
culminate from differences in the satellite products themselves (which will not be investigated here), but
also from additional sources of variability that stem primarily from differences in the criteria of input
variables. For instance, the original Westberry-CbPM study used a mixed layer definition of $\Delta T_{10m}$ = 0.5°C,
whereas the NPP products applied here use a density criteria of $\Delta\sigma_{10m}$ = 0.030 kg m$^{-3}$. If we instead derive
NPP from an MLD that is defined with a density criteria of $\Delta\sigma_{10m}$ = 0.125 kg m$^{-3}$ (as per the alternative
MLD      criterion      listed      on      the      Ocean      Productivity      website
(http://sites.science.oregonstate.edu/ocean.productivity/)) (Ryan-Keogh, 2023c) we see a further
improvement in the spatial correlation of NPP for the Westberry-CbPM (Figure A5a-d), for both SeaWIFS
($R^2$ = 0.65) and MODIS time periods ($R^2$ = 0.74) as well as the Silsbe-CAFE model for both SeaWIFS ($R^2$
= 0.82) and MODIS ($R^2$ = 0.77), with poor agreement still persisting in the equatorial Atlantic and Arabian
Sea.





Another potential source of variability for the Westberry-CbPM model specifically lies in the data source
used for determining the nitracline depth. Westberry et al. (2008) originally used the WOA01 data product
whereas here we have used the updated WOA18 product. As a brief investigation on differences between
datasets we looked at examples of the total number of nitrate data points in WOA09 and WOA13, 1186280
and 3603293 respectively, compared to WOA18, 4097914, representing increases of 203% and 14%
respectively. Further analysis investigated differences in the nitracline depth if derived using WOA13
versus WOA18 (Figure A7) results show that differences occupy the same spatial extent as the areas of
poor spatial correlation. Future versions of this product will need to incorporate updates to global nitrate
climatologies, such as the planned release of WOA23 which will greatly improve estimates of the nitracline
depth.

The remaining potential sources of variability, specific to the Silsbe-CAFE model, are the choice of salinity
data for deriving the backscattering of pure water ($b_{bw}$) and the derivation of the spectral slope of $b_{bp}$ ($\eta$).
In Silsbe et al. (2016) they assumed a constant salinity of 32.5 for simplicity, whereas here we have used
monthly means of salinity taken from WOA18. The difference between this reference value and the monthly
means (Figure A8) show that areas such as the equatorial Pacific and Atlantic, which had the lowest spatial
correlations for the Silsbe-CAFE model, have some of the biggest differences in salinity. A sensitivity
analysis of the Zhang and Hu (2009) derivation of backscattering by pure water shows that the incorrect
implementation of salinity can have significant implications on the final value (Figure A9). As such we
recommend the use of monthly climatologies, but in the future it will become necessary to account for
changing salinities, particularly in polar regions where changes in sea ice extent is resulting in freshening
(Haumann et al., 2020). One potential data product could be the climate change initiative satellite based sea
surface salinity product (Boutin et al., 2021), which has already shown strong promise of capturing
variations in salinity that match in situ measurements from both Argo floats and ships. As OC-CCI does
not release $\eta$ as a standard product we had to derive it using the Rrs data following equation 1 from Pitarch
et al. (2019). However, the wavelengths required for this derivation are 443 and 555 nm, with OC-CCI
having only 560 nm. Nevertheless, we find good agreement between MODIS derived $\eta$ and OC-CCI $\eta$
across the global ocean (Figure A10), with only a few areas in the Arctic that have very low agreement
(median $R^2$ = 0.78).

**4 Conclusion**

The data product presented here provides a continuous record of global satellite derived NPP at 8-day and
monthly resolution using multiple algorithms applied to the OC-CCI product as the longest continuing





record of satellite ocean colour (Sathyendranath et al., 2019a). The purpose is not to advocate for the
suitability of one NPP model over another, as other studies have already highlighted the strengths and
weaknesses of different satellite NPP algorithms ability to capture the appropriate range of in situ NPP
measurements (Saba et al., 2011; Friedrichs et al., 2009; Carr et al., 2006; Campbell et al., 2002). Rather,
the strength in this multi-model data product lies in its ability to offer a range of NPP across different
algorithms either as a climatology or as a long-term climatic trend for a user's specific region of interest.
Additionally, by providing multiple algorithms the user can interrogate the distribution of NPP across
different models to identify consensus or outliers that can inform decisions on whether or not to retain or
reject specific algorithms in their regional analysis. Flexibility also exists on decisions around the mixed
layer depth with two different density criteria ($\Delta\sigma_{10m} = 0.030$ or $0.125$ kg m$^{-3}$) or products (HYCOM versus
Hadley) that can be altered to ensure that the MLD input best reflects the user's region of interest. Currently
the OC-CCI is released on an annual basis with specific corrections and adjustments made based upon
assessments of previous single sensor data streams and any new data sources. The multi-model data product
presented here will be updated on the same regular basis as and when OC-CCI data is updated, with
backwards corrections similarly applied to prevent the retention of erroneous values in the data record.
Future updates to this data product will similarly incorporate not only updated climatological mean values
(i.e., the planned release of WOA2023), but will also incorporate additional NPP algorithms, (i.e., SABPM;
Tao et al., 2017). to provide the user with a wide range of options for assessing climatological seasonal
cycles as well as trends and trajectories of oceanic productivity.
**Appendices**



Earth System
Science
Data

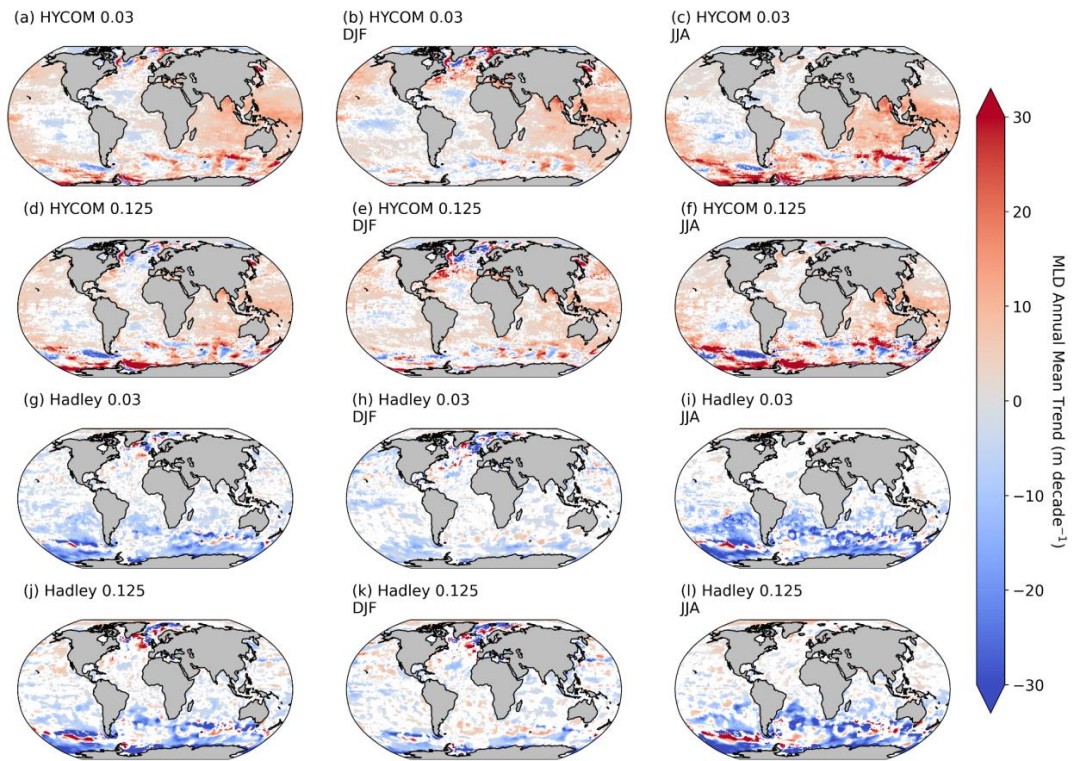

Figure A1: The annual mean trends of the different MLD data products HYCOM (a-f) and Hadley (g-l) for the different criterion of $\Delta\sigma_{10m} = 0.030$ kg m$^{-3}$ (a-c,g-i) and $\Delta\sigma_{10m} = 0.125$ kg m$^{-3}$ (d-f,j-l) averaged for the whole year (a,d,g,j), December to February (b,e,h,h) and June to August (c,f,i,l). Trend analysis performed as described in Ryan-Keogh et al. (2023).

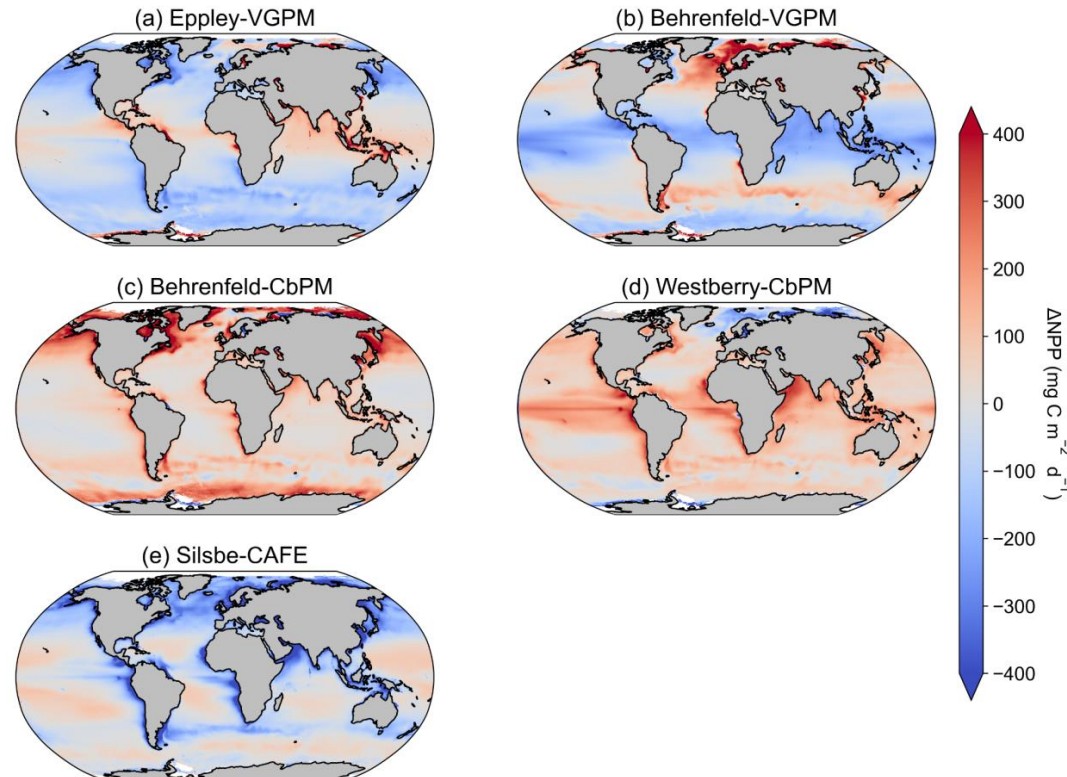

Figure A2: The difference in climatological mean [1998-2022] NPP between the inter-model mean and (a) Eppley-VGPM, (b) Behrenfeld-VGPM, (c) Behrenfeld-CbPM, (d) Westberry-CbPM and (e) Silsbe-CAFE models.

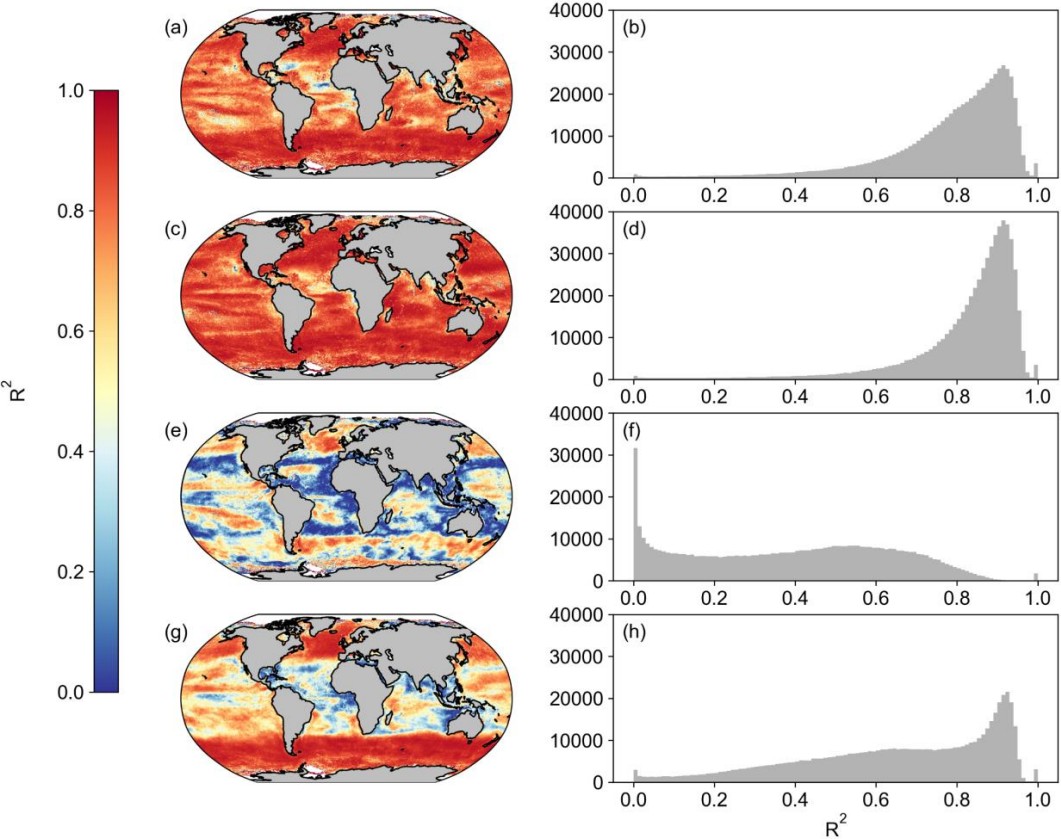

434

Figure A3: Spatial correlation maps and histograms of Pearson's correlation coefficient $R^2$ values between SeaWIFS and OC-CCI for the period of 1998-01-01 to 2007-12-31 for (a,b) Eppley-VGPM, (c,d) Behrenfeld-VGPM, (e,f) Westberry-CbPM and (g,h) Silsbe-CAFE. Please note that for Westberry-CbPM and Silsbe-CAFE, the MLD product used for SeaWIFS is HYCOM and the MLD product for OC-CCI is Hadley, both using the $\Delta\sigma_{10m} = 0.030$ kg m$^{-3}$ criterion.

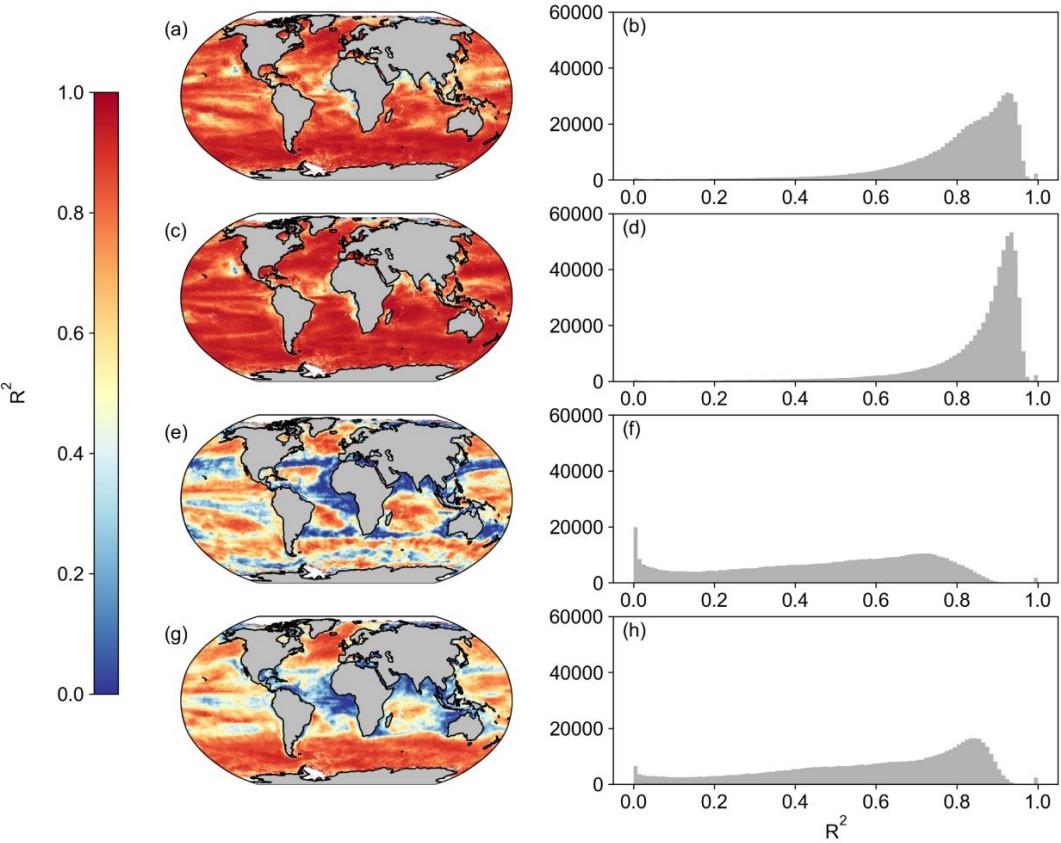

440

Figure A4: Spatial correlation maps and histograms of Pearson's correlation coefficient $R^2$ values between
MODIS and OC-CCI for the period of 2003-01-01 to 2019-12-31 for (a,b) Eppley-VGPM, (c,d) Behrenfeld-
VGPM, (e,f) Westberry-CbPM and (g,h) Silsbe-CAFE. Please note that for Westberry-CbPM and Silsbe-
CAFE, the MLD product used for SeaWIFS is HYCOM and the MLD product for OC-CCI is Hadley, both
using the $\Delta\sigma_{10m} = 0.030$ kg m$^{-3}$ criterion.



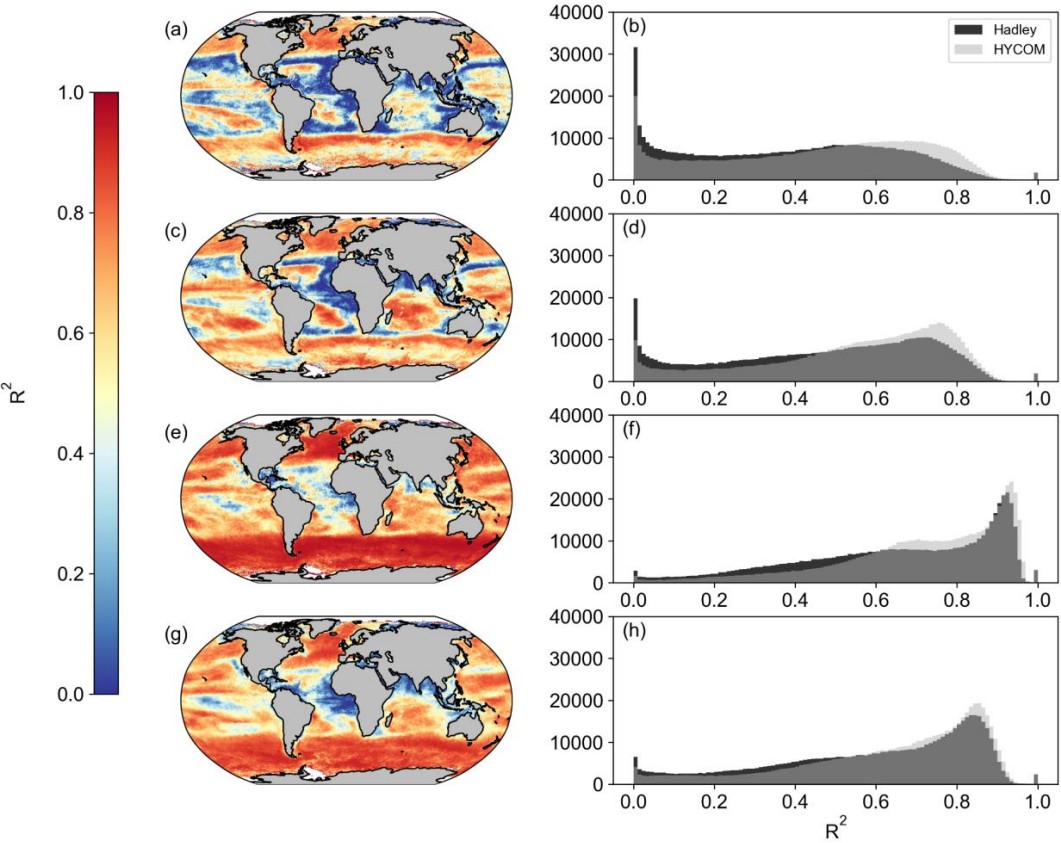

Figure A5: Spatial correlation maps and histograms of Pearson's correlation coefficient $R^2$ values between SeaWIFS (a,b,e,f), MODIS (c,d,g,h) and OC-CCI for (a,b,c,d) Westberry-CbPM and (e,f,g,h) Silsbe-CAFE. Please note that the MLD product used is HYCOM with the $\Delta\sigma_{10m} = 0.030$ kg m$^{-3}$ criterion. Included in the histograms are the Pearson's correlation coefficient $R^2$ values using the Hadley MLD data product (in black) as displayed in Figures A3 and A4.

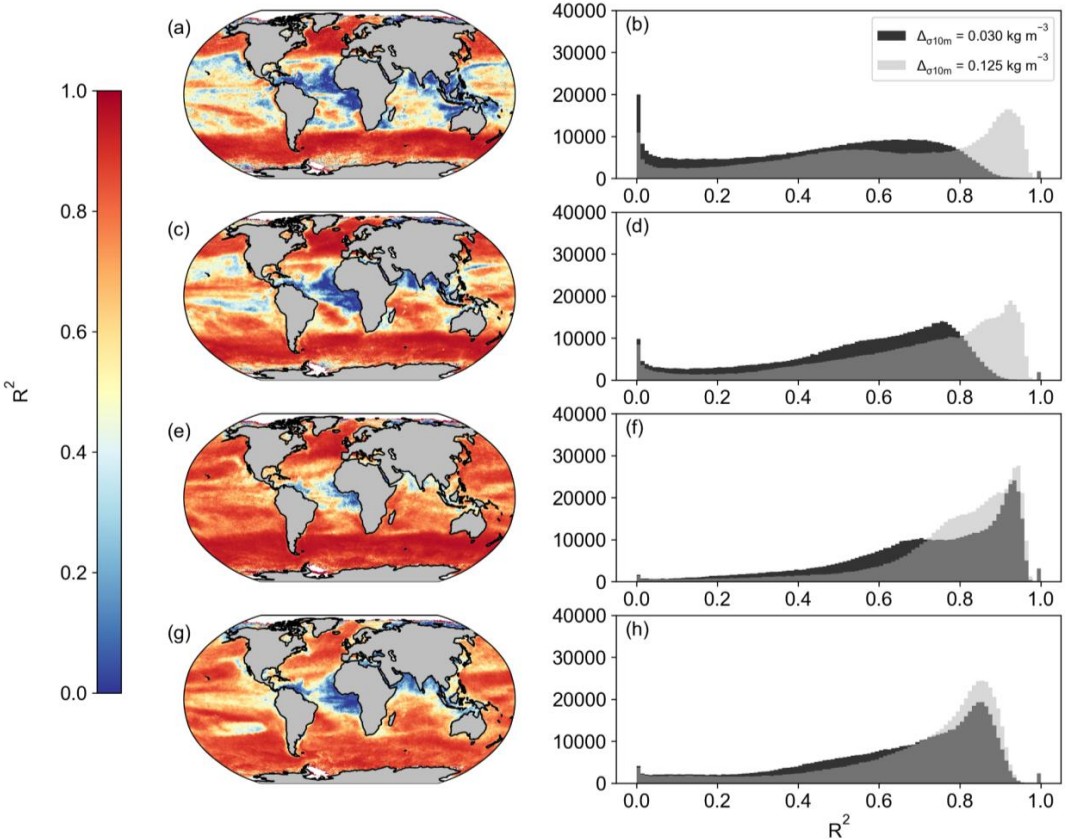


Figure A6: Spatial correlation maps and histograms of Pearson's correlation coefficient $R^2$ values using the
MLD criterion of $\Delta\sigma_{10m} = 0.125$ kg m$^{-3}$ (in grey) for (a,b) Westberry-CbPM SeaWIFS vs OC-CCI, (c,d)
Westberry-CbPM MODIS vs OC-CCI, (e,f) Silsbe-CAFE SeaWIFS vs OC-CCI and (a,b) CAFE MODIS
vs OC-CCI. Included in the histograms are the Pearson's correlation coefficient $R^2$ values using the MLD
criterion of $\Delta\sigma_{10m} = 0.030$ kg m$^{-3}$ (in black) as displayed in Figures A3 and A4.

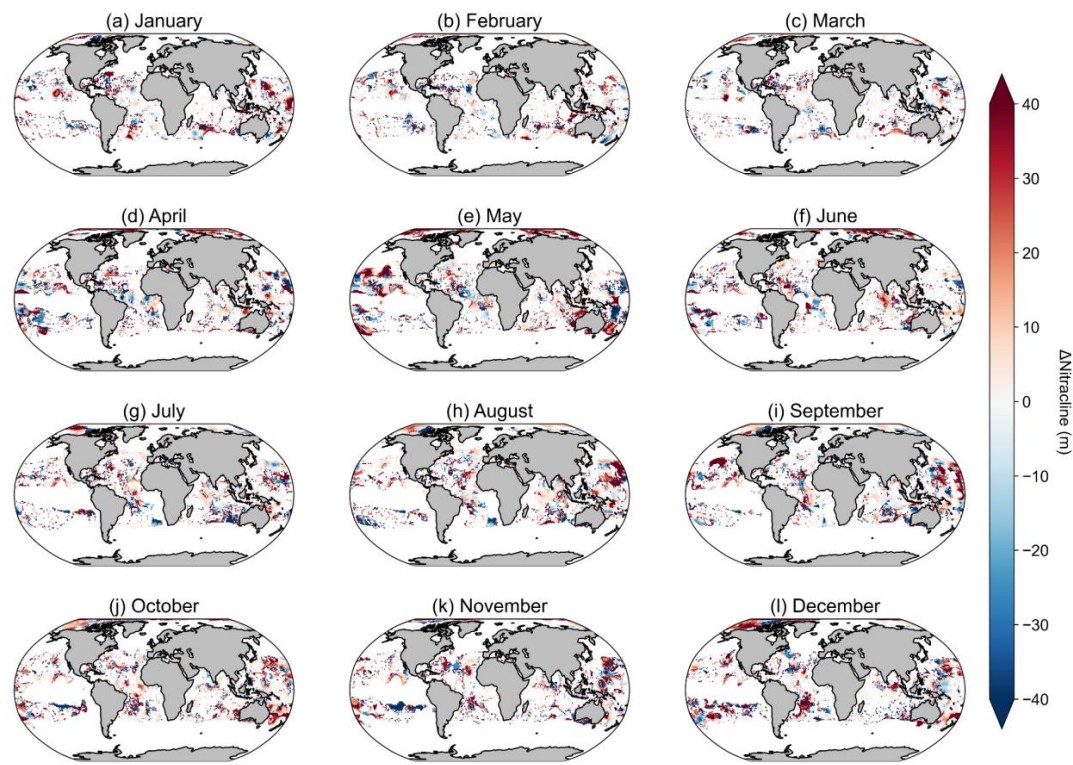

Figure A7: Maps of the difference in nitracline depth, where the nitracline depth is calculated as the depth
at which nitrate + nitrite is equal to 0.5 μM, between monthly WOA2013 and WOA2018.



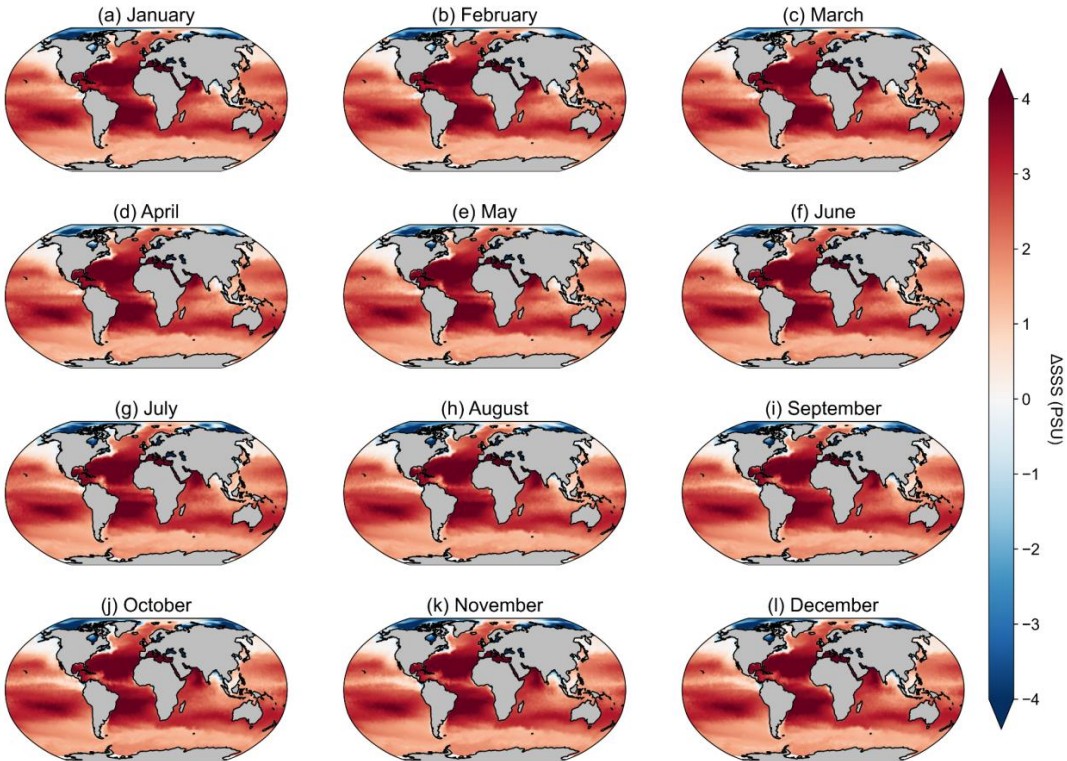


Figure A8: Maps of the difference in sea surface salinity (SSS) from the WOA18 monthly climatology and
the reference SSS value used in Silsbe et al. (2016) of 32.5 PSU.



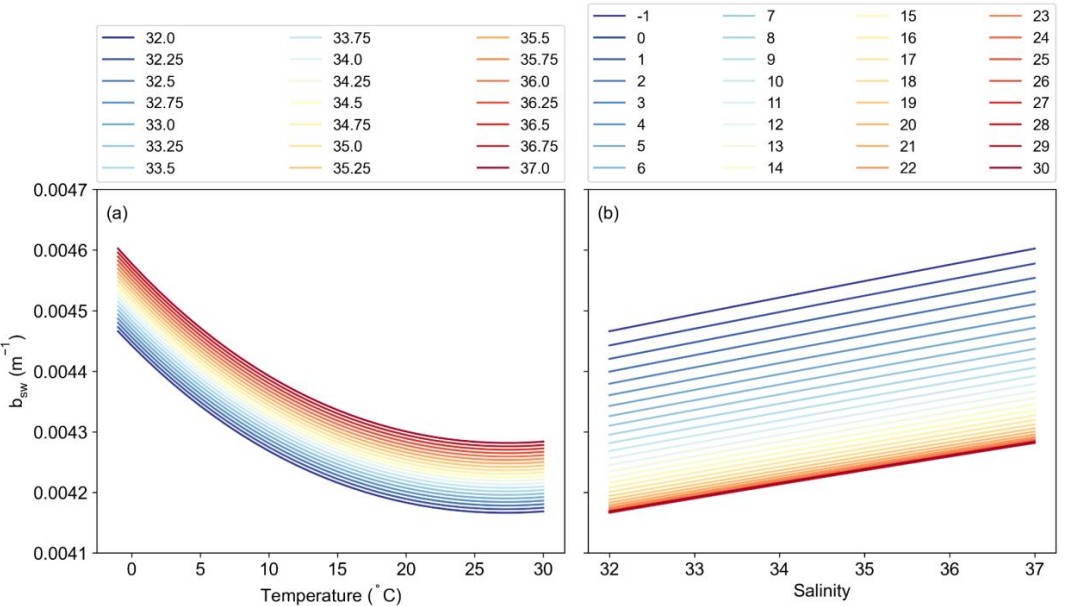

Figure A9: Sensitivity analysis of the calculation of the total backscattering of pure seawater ($b_{sw}$; $m^{-1}$) as a function of both (a) Temperature (°C) (colour scale = Salinity) and (b) Salinity (colour scale = Temperature (°C)).




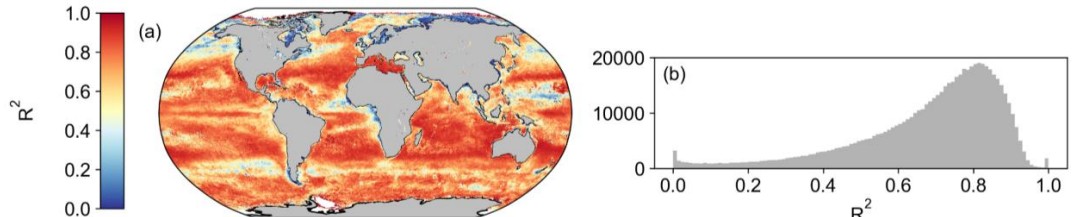


Figure A10: A spatial correlation map (a) and a histogram of Pearson's correlation coefficient $R^2$ values (b)
between monthly MODIS and OC-CCI derived spectral slope of $b_{bp}$ ($\eta$) for the period of 2003-01-01 to

473    2019-12-31.


**Data Availability**

The primary manuscript data are available at: https://doi.org/10.5281/zenodo.7849935 (Ryan-Keogh,
2023d). The NPP products which used Hadley $\Delta\sigma_{10m}$ = 0.125 kg m$^{-3}$ data are available at:
https://doi.org/10.5281/zenodo.7858590 (Ryan-Keogh, 2023a). The NPP products which used HYCOM
$\Delta\sigma_{10m}$ = 0.030 kg m$^{-3}$ data are available at: https://doi.org/10.5281/zenodo.7860491 (Ryan-Keogh, 2023b).
The NPP products which used HYCOM $\Delta\sigma_{10m}$ = 0.125 kg m$^{-3}$ data are available at:
https://doi.org/10.5281/zenodo.7861158 (Ryan-Keogh, 2023c). OC-CCI data were downloaded from
https://www.oceancolour.org/. SeaWIFS and MODIS NPP data products used for the comparison were
downloaded           from           the           Ocean           Productivity           website
(http://sites.science.oregonstate.edu/ocean.productivity/). The Hadley gridded temperature and salinity data
were downloaded from https://www.metoffice.gov.uk/hadobs/en4/. The HYCOM MLD data were
downloaded           from           the           Ocean           Productivity           website
(http://sites.science.oregonstate.edu/ocean.productivity/).     PAR     data     were     downloaded     from
http://www.globcolour.info/. Sea surface temperature data were downloaded from https://www.ghrsst.org/.

**Author Contribution**

Conceptualization: TJRK, SJT
Formal Analysis: TJRK
Methodology: TJRK
Software: TJRK, NC, TM
Visualisation: TJRK





Writing - original draft: TJRK
Writing - reviewing & editing: TJRK, SJT, NC, TM

**Competing Interests**

The authors declare no competing interests.

**Acknowledgements**
We would like to acknowledge the OC-CCI group for providing the satellite data used in this manuscript.
The authors acknowledge their institutional support from the CSIR Parliamentary Grant and the Department
of Science and Innovation. We similarly acknowledge the Centre for High-Performance Computing (CSIR-
CHPC) for the support and computational hours required for the analysis of this work. We would like to
acknowledge Greg Silsbe for sharing the code for the CAFE algorithm.

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
