# Peer review of "A new global oceanic multi-model net primary productivity data product"

_Earth System Science Data, 2023_

## Referee Comment (RC2)

Review of "*A new global oceanic multi-model net primary productivity data product*", by Ryan-Keogh et al.

This manuscript introduces a new data product that consists of an ensemble mean and associated variances derived from a suite of diverse global satellite net primary production (NPP) models. The data product is generated by application of the various NPP models to an established, merged multi-mission ocean color record that spans the full modern satellite record (1998-2022). The manuscript describes the approach and some of the basic spatio-temporal patterns observed in the product. It is well-written and the graphics are good quality. I recommend this manuscript for publication without major changes. It does not introduce any new science per se, but provides a product that will hopefully be of use to the broader science community. I offer the following points out of general interest and perhaps to better-clarify certain points.

First, the authors acknowledge the need for this product to provide an alternative to other data products already in existence (e.g., those hosted by Copernicus Marine Services or the Oregon State University Ocean Productivity website). However, in doing so, I feel that a massive disclaimer is needed stating that advantages of this 'ensemble approach' may be fully mitigated by combination of estimates of varying quality (as assessed by exercises such as the Primary Productivity Algorithm Round Robin series, Campbell et al. (2002); Carr et al., (2006); Friedrichs et al. (2009); Saba et al. (2011); Lee et al. (2015)). While exercises such as these are not definitive, the community has dedicated tremendous effort to trying to establish the fidelity of these models. This has been challenging and is limited partly by satellite – in situ matchups, but I feel that we were at least converging on a narrative that CAFE>CbPM>VGPM. Personally, I would gladly use the CAFE model applied the OC-CCI dataset as the preferred data product (but I realize not everyone may feel the same :). Perhaps, a bit of a philosophical point, but I think it deserves some discussion. It could easily be placed in the paragraph starting on Line 96 to balance the justification for the current product.

I'm concerned about the global annual integrated NPP values. They are really high, much moreso than we have reported in the original publications and related work. I wonder if the integrals are unduly influenced by a low number of 'spurious' values? They might be easily traceable to something in the input fields (e.g., coastal Chl-a retrievals > 50 mg/m3, or spurious bbp retreivals in the case of the CbPM). It would take more than a pixel or two, but implementing these 'traps' on the input, as well as the resultant NPP, can be important. Also, in the case of the Behrenfeld-CbPM, the formulation inappropriately uses Kd490 to estimate the euphotic depths and mixed layer growth irradiances, both of which can be significantly overestimated in this way. In Westberry et al. (2008), we point out that the global annual NPP is reduced by nearly 2x (from 67 to 35 Pg C) by simply replacing the Kd490 terms with simple Chl-dependent parameterizations (e.g., Morel-type relationships).

Figure 2a, I think you should truncate the CV map at 0.6 or so I order to see more spatial structure? Just a thought ...

Lines 262-265, the convergence or divergence of model NPP over time is interesting.  related to the inter-sensor merging of the OC-CCI record.  This could be investigated by looking at single missions, with MODIS-Aqua being the obvious candidate because of the record length.

Line 238, typo should read 'Westberry-CBPM'

---

## Author Comment (AC1)

**Reviewer 1**

Revision of Ryan-Keogh et al. "A new global oceanic multi-model net primary productivity data product"

General comment
The paper submitted by Ryan-Keogh et al. focus on satellite NPP estimates based on the existing algorithms. To this aim, the authors have used the longest satellite time-series available from the ESA OC CCI project. The paper is well written and easy to be read and it is a useful work for the ocean colour community.
Overall, I have few specific comments to be included in the revised version of the manuscript. For instance, one point to be addressed is about the differences between the NPP products based on ESA OC CCI dataset and the NPP estimates for MODIS and SeaWiFS. I suggest to extend this paragraph also including some lines about the difference ocean color bio-optical algorithms used in the NPP computations. For instance, the use of GSM instead of QAA that is used in OC-CCI to retrieve the backscatter; this is valid also for the chlorophyll concentration estimates.
On my opinion, the paper needs only minor corrections before the publication.

> We would like to thank the reviewer for taking the time to review our manuscript and we hope that the changes we have made are suitable. For responses to the specific comments please see below where we have amended the text to either rephrase or update sections as noted.

Specific comments:

Lines 26-27: I'd suggest to rephrase the sentence and include a more generic concept about the spatial and temporal variability of phytoplankton dynamics instead of only the seasonal bloom as a key biological process which play an important role in the carbon.

> We thank you for this suggestion and have amended the sentence as follows.

> "Phytoplankton primary production and associated spatial and temporal variability play an important role in the carbon cycle, being responsible for approximately 50% of total global net primary production (NPP)"

Lines 30-36: For me this sentence is not clear, please could you rephrase. There are also some repetions that need to be removed.

> We have revised the sentences to now read as follows.

> "When this organic carbon is sequestered to the ocean interior via the biological carbon pump (BCP) it offsets the flux of upwelled pre-industrial dissolved inorganic carbon (DIC) (Mikaloff Fletcher et al., 2007; Gruber et al., 2009), where DIC is the carbon source for phytoplankton photosynthesis. In that sense, in the contemporary period, it does not play a

significant role in the ocean uptake of anthropogenic carbon dioxide ($CO_2$). However, the magnitude of the BCP is predicted to change in response to global climate change, which will alter the ocean's ability to store carbon and therefore impact atmospheric levels of $CO_2$ (Henson et al., 2011; Bopp et al., 2013; Boyd et al., 2015; Tagliabue et al., 2021). Such changes are of concern because alterations in the contribution that the BCP plays in offsetting upwelled DIC will impact the net uptake of anthropogenic $CO_2$ (Henson et al., 2011). As such any natural or anthropogenic perturbations to the strength and efficiency of the BCP have the potential to drive important feedbacks on global climate change and thus need to be considered for a comprehensive understanding of the trajectory of the ocean carbon sink."

Lines 40-41: Maybe you can listed areas where there is an increase and areas where there is a decrase of NPP.

We thank you for this suggestion and have now amended the sentence as follows.

"Recent studies have estimated that global NPP is indeed changing, with declines ranging from 0.6 to 13% across equatorial and temperate regions (Gregg and Rousseaux, 2019; Polovina et al., 2011; Chavez et al., 2010; Behrenfeld et al., 2006) and increases of up to 2% at the Bermuda Atlantic Time Series and Hawaii Ocean Time Series (Saba et al., 2010)."

Lines 42: DIC is the first time cited. Please include the acronyms. In addition, I suggest to include just a sentence to introduce the role and importance of DIC and its connection with the biological carbon pump.

We have amended it so that when dissolved inorganic carbon is introduced in line 32 we have put the acronym DIC afterwards. Please amended sentence below.

"When this organic carbon is sequestered to the ocean interior via the biological carbon pump (BCP) it offsets the flux of upwelled pre-industrial dissolved inorganic carbon (DIC) (Mikaloff Fletcher et al., 2007; Gruber et al., 2009), where DIC is the carbon source for phytoplankton photosynthesis."

Lines 43-47: This paragraph can be reduced in terms of lines. An option should be to refer directly to the trophic chain or food web.

We thank you for this suggestion. We have moved the following sentence on the impacts of a changing BCP up to where it was introduced originally.

"These changes are of concern given that alterations in the contribution that the BCP plays in offsetting upwelling of DIC will impact the net uptake of anthropogenic $CO_2$ (Henson et al., 2011)."

Lines 55-57: In which sense the NPP is a better proxy environmental changes? It is because the phytoplankton chlorophyll is also influenced by physiological adaptations to

light, nutrient and temperature? Maybe you can add a short sentence and a reference to better clarify this point.

> We have added some additional details to this section with the reasoning highlighted in Tilstone et al. (2023). As such these sentences now read.
>
> "NPP has already been highlighted as a better indicator of environmental change and disturbances in comparison to chlorophyll-a (Tilstone et al., 2023), with environmental disturbances (i.e., changes in nutrient inputs) being detected through changes in phytoplankton photosynthetic rates and NPP (Boalch, 1987), highlighting its suitability for ecosystem assessment of tipping points and abrupt change."

Line 60: Nutrient availability instead of macro- and micro-nutrient specification.

> Thank you for the suggestion, we have amended the sentence as follows.
> Line 61: "Phytoplankton NPP is strongly influenced by the physico-chemical conditions of the ocean, including light, temperature and nutrient availability."

Lines 66-68: You should include a reference at the end of the sentence. For instance Johnson and Bif (2021; Nature Geoscience).

> We thank you for this suggestion and have added the reference for Johnson and Bif (2021) at the end of the sentence.

Lines 87-89: You should include also that this programme is funded by th European Space Agency (ESA). Maybe some lines about the programme. The main goal of the OC-CCI is to have a long-term consistency merged ocean color dataset. OC-CCI data products are the result of the merging of SeaWiFS, MERIS, MODIS, and VIIRS observations in which the inter-sensor biases are removed. version 6.0 includes an updated processing, which mostly accounts for the ageing of the MODIS sensor and the new sensors (Sentinel-3/OLCI).

> We thank you for these suggestions and in the first instance when OC-CCI is mentioned we have included the European Space Agency. This section now reads:
>
> "Recently, considerable effort has been invested by the European Space Agency to provide one of the longest records of ocean colour for detecting climate variability by merging data, from SeaWIFS, MODIS, MERIS, VIIRS, Sentinel 3A OLCI and Sentinel 3B OLCI, and correcting inter-sensor biases from multiple ocean colour satellite sensors (Sathyendranath et al., 2019a), known as the Ocean Colour Climate Change Initiative (OC-CCI)."
>
> We decided rather than to put additional details here about v6.0 we have instead included it in the methods.
>
> "25 years of ocean colour data from 1998 – 2022 were downloaded from the OC-CCI server (8-day; 4 km; v6.0; Sathyendranath et al., 2019), in which the latest version v6.0

includes an updated MERIS-4$^{th}$ reprocessing, the inclusion of Sentinel 3B OLCI, the dropping of MODIS and VIIRS data after 2019 and the use of the Quasi-Analytical algorithm (QAA) (Lee et al., 2002)."

Lines 97-99: Maybe, you should say like: NPP from OC-CCI is not available yet since it is not a standard product of the programme. Kulk et al., (2020) have used the dataset to develop and validate an algorithm that however is not listed in the suite of variables of the OC-CCI.

We thank you for this suggestion, however the next sentence already includes a statement to this effect.

"Unfortunately, the global NPP algorithm applied to OC-CCI by Kulk et al. (Kulk et al., 2020) is not available for download on the OC-CCI server."

Lines 105-106: Probably you should smooth the sentence a little bit as: however, it is difficult to obtain all the variables needed for the detection of NPP such as the mixed layer depth, the nitracline, ecc

Perhaps we did not make this part as clear as possible. We are not raising the issue that it is difficult for the user to obtain these ancillary variables, but rather that there is a lack of information provided by the Ocean Productivity Group over the details of the variables used in each algorithm. For example, on their website you can download HYCOM MLD data with 2 different criteria. Yet, when you access the NPP data there is only 1 version available with no metadata provided in the file to inform you of which MLD criteria was used.

We have tried to clarify this sentence as follows:

"Furthermore, it is difficult for the user to ascertain exactly which ancillary data products (i.e., MLD criterion, nitracline) were used in the empirical derivations of the single sensor NPP products available for download."

Lines 122-123: The spectral slope is not a product listed in the OC-CCI project, so you should say: we have computed the spectral slope of bbp (xx) following.

We are unsure what you are referring to as we have the following statement on line 125-127.
"As the OC-CCI server does not contain the spectral slope of b$_{bp}$ ($\eta$; m$^{-1}$ nm$^{-1}$), it was calculated following equation 1 from Pitarch et al. (2019) using remote sensing reflectance (R$_{rs}$) at 443 nm and 560 nm."

To provide greater clarity we have amended the sentence as follows:

"As the spectral slope of $b_{bp}$ ($\eta$; $m^{-1}$ $nm^{-1}$) is not a variable provided by the OC-CCI project it had to be calculated following equation 1 from Pitarch et al. (2019) using remote sensing reflectance ($R_{rs}$) at 443 nm and 560 nm."

Line 129: Maybe a reference also for the criteria of 0.125 kg/m3

We have added the following reference for the criterion of 0.125 $kg/m^3$ - Suga et al. 2004, Journal of Physical Oceanography, doi: https://doi.org/10.1175/1520-0485(2004)034<0003:TNPCOW>2.0

Line 145: Did you use monthly climatology data or monthly data for the entire time-series?

The world ocean atlas is the monthly climatology of nitrate profiles, we have amended the sentence to make this clearer:

"The nitracline depth was defined as the depth at which nitrate and nitrite was equal to 0.5 µM (Westberry et al., 2008), using the monthly climatology nitrate and nitrite profile data from the World Ocean Atlas 2018 (WOA18; (Garcia et al., 2019)."

Lines 180-185: You should start as: "As additional term of comparison, monthly NPP data of.

Thank you for this suggestion we have amended the sentence to read as follows:

"As an additional comparison to the OC-CCI outputs presented here, monthly NPP data of"

Lines 397: I suggest also to include the algorithms at the base of the ocean color quantities such as chlorophyll, backscatter ecc. For instance the chlorophyll in ESA OC-CCI dataset is based on different algorithms as well as the backscattering coefficient is based on the QAA algorithm. The Behrenfeld NPP CbPM is based on the GSM that sometimes overetimates backscattering values at 443. For this reason, you should extend the last paragraph of the results and discussion taking into account this additional source of differences (algorithms).

We thank you for this suggestion to expand the results and discussions to include the sources of variability derived from the choice of IOP algorithm. This section now includes the following additional discussion.

"The differences highlighted here in $\eta$ and the inherent optical properties (IOP), which are required for the derivation of each NPP model, can be explained by the use of different ocean colour algorithms. For example, OC-CCI uses QAA that requires multiple $R_{rs}$ bands (typically 6 or more) and can account for variability in the spectral shape of reflectance and the IOPs (i.e., $b_{bp}$, $a_{ph}$, $a_{dg}$). This makes this algorithm suitable for multiple water types from the open ocean to optically complex coastal waters. A different algorithm which is typically used to process data from MODIS is the Garvel-Siegel-Maritorena (GSM)

(Garver and Siegel, 1997; Maritorena et al., 2002) algorithm that only requires 3 $R_{rs}$ bands, and therefore does take into account spectral variability meaning it is typically only suited for open ocean waters. Indeed, some studies have highlighted how the GSM model can sometimes overestimate $b_{bp}$ ($\lambda$443) values (Brewin et al. 2015), which would directly impact the NPP algorithms here which use this IOP to estimate phytoplankton carbon."

**Reviewer 2**

Review of "*A new global oceanic multi-model net primary productivity data product*", by Ryan-Keogh et al.

This manuscript introduces a new data product that consists of an ensemble mean and associated variances derived from a suite of diverse global satellite net primary production (NPP) models. The data product is generated by application of the various NPP models to an established, merged multi-mission ocean color record that spans the full modern satellite record (1998-2022). The manuscript describes the approach and some of the basic spatio-temporal patterns observed in the product. It is well-written and the graphics are good quality. I recommend this manuscript for publication without major changes. It does not introduce any new science per se, but provides a product that will hopefully be of use to the broader science community. I offer the following points out of general interest and perhaps to better-clarify certain points.

> We thank the reviewer for their suggestions and comments on how to improve this manuscript. We have taken these on board and amended the manuscript as suggested. We hope that the reviewer finds these changes suitable and that the manuscript is now improved.

First, the authors acknowledge the need for this product to provide an alternative to other data products already in existence (e.g., those hosted by Copernicus Marine Services or the Oregon State University Ocean Productivity website). However, in doing so, I feel that a massive disclaimer is needed stating that advantages of this 'ensemble approach' may be fully mitigated by combination of estimates of varying quality (as assessed by exercises such as the Primary Productivity Algorithm Round Robin series, Campbell et al. (2002); Carr et al., (2006); Friedrichs et al. (2009); Saba et al. (2011); Lee et al. (2015)). While exercises such as these are not definitive, the community has dedicated tremendous effort to trying to establish the fidelity of these models. This has been challenging and is limited partly by satellite – in situ matchups, but I feel that we were at least converging on a narrative that CAFE>CbPM>VGPM. Personally, I would gladly use the CAFE model applied the OC-CCI dataset as the preferred data product (but I realize not everyone may feel the same :). Perhaps, a bit of a philosophical point, but I think it deserves some discussion. It could easily be placed in the paragraph starting on Line 96 to balance the justification for the current product.

> We thank the reviewer for this suggestion, and indeed it is the personal opinion of the authors that the CAFE model is definitely one of the best NPP models available for use given its optical complexity for handling energy absorption whilst also accounting for diurnal variation in light availability. We are cognisant though that in the community there are still debates over which model is best, even after the round robin exercises (which in Silsbe et al. 2015 did highlight the CAFE model as having the lowest bias and error). Rather than try to influence the community over which model is best we wished to provide them with a range of options for them to make an informed choice. For example, in a recent paper we published in Science we utilised the Behrenfeld-VGPM, Behrenfeld-CbPM,

Westberry-CbPM and Silsbe-CAFE to look at trends in NPP in the Southern Ocean. Where the models agreed in the trend direction we had confidence that NPP is most likely declining, but when the models disagreed (i.e., the Behrenfeld-VGPM showed increases) we had to find a plausible explanation for this. We note however the value and importance of PPARR exercises and have included the following statement to this effect:

"It is worth noting however, that previous studies have performed a series of statistical evaluations of NPP models, known as Primary Production Algorithm Round Robin (Campbell et al., 2002; Carr et al., 2006; Friedrichs et al., 2009; Saba et al., 2011; Lee et al., 2015), utilising in situ measurements and satellite matchups to assess their relative performance, with most recently the CAFE model having the lowest bias and error in comparison to all other algorithms available at the time."

I'm concerned about the global annual integrated NPP values. They are really high, much moreso than we have reported in the original publications and related work. I wonder if the integrals are unduly influenced by a low number of 'spurious' values? They might be easily traceable to something in the input fields (e.g., coastal Chl-a retrievals > 50 mg/m3, or spurious bbp retrievals in the case of the CbPM). It would take more than a pixel or two, but implementing these 'traps' on the input, as well as the resultant NPP, can be important. Also, in the case of the Behrenfeld-CbPM, the formulation inappropriately uses Kd490 to estimate the euphotic depths and mixed layer growth irradiances, both of which can be significantly overestimated in this way. In Westberry et al. (2008), we point out that the global annual NPP is reduced by nearly 2x (from 67 to 35 Pg C) by simply replacing the Kd490 terms with simple Chl-dependent parameterizations (e.g., Morel-type relationships).

We thank the reviewer for raising this concern and indeed we did find a small issue with the CbPM models where the code for replacing $b_{bp}(\lambda 443)$ values below 0.0035 was also replacing pixels where no data was recorded. We have since reran the code for every algorithm from start to finish and recalculated all of the values in Table 2 and replotted all figures. It is worth noting that for our estimates of global oceanic NPP there was another error in accounting for the size of pixels at the equator in comparison to pixels at higher latitudes. We used the description of how to calculate global oceanic NPP taken from the Ocean Productivity Website to design a new workflow for calculating these values. With the reprocessing and new formulation the range of values for all models now sits between 46.4 and 66.2 Pg C m$^{-2}$ yr$^{-1}$, in comparison to the previous range of 58.9 to 87.7 Pg C m$^{-2}$ yr$^{-1}$, which is more inline with those reported in the original publications and related work.

Figure 2a, I think you should truncate the CV map at 0.6 or so I order to see more spatial structure? Just a thought …

We thank you for this suggestion, we have changed the colorbar scale to be 0.1 to 0.5, and we can see more spatial structures in the coefficient of variation. Please see image below for the changes.

[Figure]

Lines 262-265, The convergence or divergence of model NPP over time is interesting. Is it related to the inter-sensor merging of the OC-CCI record? Is it related to the fact that some models utilize products that others do not (e.g., MLD)? Resolving this is beyond the scope of this paper, but to me, this is related to the point above regarding the advantages and disadvantages of the ensemble mean versus a single model that we believe performs best.

We thank you for raising this interesting point and we do agree that investigating why models may diverge or converge in different areas of the ocean does raise further questions. Our ensemble approach here we believe helps to highlight to the user that whilst all NPP models have unique strengths and weaknesses, these only become apparent when in comparison to all other models. If we consider a scenario where 4 out of 5 models are predicting a decreasing trend in NPP for a region (or a similar climatology), with 1 model predicting an increasing trend (or an inconsistent climatology), then it is most likely that NPP is decreasing and the user can have confidence in using any one of the three models that are in agreement. If however the user only had access to 1 model applied to 1 single sensor mission then they may not have as much confidence in the data set to make robust

conclusions on the trends or climatologies. We hope that this data product will provide this opportunity for the community to begin to investigate these sorts of changes in oceanic NPP. Even if one model is typically better than another, it is not necessarily the best always and everywhere (e.g. in some regions and seasons it makes little difference which model is used and in other instances it makes a big difference). Resolving the "why's" of these discrepancies were indeed beyond the scope of this model but we did want to try to make it clear to users that they should investigate all models and determine which model or combination of models is best for their region or study.

Line 238, typo should read 'Westberry-CBPM'

We thank you for finding this typo, we have since corrected it to read Westberry-CbPM.